# Depth and benthic habitat influence shallow and mesophotic predatory fishes on a remote, high-latitude coral reef

Kristy Brown[1], Jacquomo Monk[1]*, Joel Williams[1,2], Andrew Carroll[3], David Harasti[2], Neville Barrett[1]

1 Institute for Marine and Antarctic Studies, University of Tasmania, Hobart, Tasmania, Australia, 2 Fisheries Research, Port Stephens Fisheries Institute, NSW Department of Primary Industries, Taylors Beach, NSW, Australia, 3 Geoscience Australia, Canberra, ACT, Australia

* jacquomo.monk@utas.edu.au

## Abstract

Predatory fishes on coral reefs continue to decline globally despite playing key roles in ecosystem functioning. Remote atolls and platform reefs provide potential refugia for predator populations, but quantitative information on their spatial distribution is required to establish accurate baselines for ongoing monitoring and conservation management. Current knowledge of predatory fish populations has been derived from targeted shallow diver-based surveys (<15 m). However, the spatial distribution and extent of predatory fishes on outer mesophotic shelf environments has remained under described. Middleton Reef is a remote, high-latitude, oceanic platform reef that is located within a no-take area in the Lord Howe Marine Park off eastern Australia. Here we used baited remote underwater stereo video to sample predatory fishes across lagoon and outer shelf habitats from depths 0–100 m, extending knowledge on use of mesophotic depths and habitats. Many predatory fish demonstrated clear depth and habitat associations over this depth range. Carcharhinid sharks and Carangid fishes were the most abundant predators sampled on Middleton Reef, with five predatory fishes accounting for over 90% of the predator fish biomass. Notably, Galapagos shark (*Carcharhinus galapagensis*) and the protected black rockcod (*Epinephelus daemelii*) dominated the predator fish assemblage. A higher richness of predator fish species was sampled on reef areas north and south of the lagoon. The more exposed southern aspect of the reef supported a different suite of predator fish across mesophotic habitats relative to the assemblage recorded in the north and lagoonal habitats, a pattern potentially driven by differences in hard coral cover. Biomass of predatory fishes in the more sheltered north habitats was twice that of other areas, predominantly driven by high abundances of Galapagos shark. This work adds to the growing body of literature highlighting the conservation value of isolated oceanic reefs and the need to ensure that lagoon, shallow and mesophotic habitats in these systems are adequately protected, as they support vulnerable ecologically and economically important predator fish assemblages.

**Data Availability Statement:** The data used in the paper can be accessed at: https://globalarchive.org/geodata/explore/?filters=%7B"deployment_campaign_list":%5B993%5D%7D.

**Funding:** This work was funded through the Australian Government's National Environmental Science Program (NESP) and Parks Australia.

**Competing interests:** The authors have declared that no competing interests exist. The funders had no role in study design, data collection and analysis, decision to publish, or preparation of the manuscript.

# 1 Introduction

In terrestrial and marine ecological systems, top-order predators act to shape trophic structures below them [1–3]. Coral reefs provide ecosystem services to top predators, and predatory fishes play an important role in overall ecosystem function and health [4,5]. Predator species influence prey behaviour and remove prey items from ecosystems, regulating the composition of, and dynamics within prey assemblages [6,7]. Globally, the removal of predators through overfishing can destabilise food webs through mesopredator release and herbivore suppression that leads to altered trophic function [8–10]. Knowledge of the effects of predator removal and trophic ecology on reefs remains a contemporary issue due to its conservation implications [11]. Baseline understanding of predator spatial distribution and species life history proves necessary in a paradigm where optimisation of conservation outcomes necessitates ground truthing of ecosystem dynamics [12].

Research undertaken on continental shelves suggests that predatory fish assemblages can vary considerably between, and within, habitats and depths [13–15]. Differences in physical habitat (e.g. type, structure) and oceanographic conditions (e.g. water movement, light availability and physical orientation) shape the composition of benthic biota and associated fish assemblages [16]. For example, predatory reef fishes are often highly abundant on outer reef slopes [8], where planktivorous prey aggregate due to enhanced primary productivity associated with oceanic currents [17]. Fish assemblages are also often ecologically distinct on mesophotic reefs (i.e. >30 m), where studies suggest a low abundance of herbivores, high abundance of planktivorous fishes and a concentration of predatory fish biomass [14,18]. Other studies suggest that patterns in fish assemblages (particularly predatory fishes) are shaped by a complicated synergistic combination of physical habitat characteristics, oceanographic conditions and fishing pressure (e.g. [19,20]).

Remote coral atolls and near surface seamounts act as potential refugia for top-order predatory fishes, providing respite from fishing pressure due to their geographic isolation from humans [21–23]. However, research on these remote coral reef atolls has largely been dominated by diver-based underwater visual census (UVC) and has focused on the shallower (0–20 m) margins of these systems [24,25]. These remote reefs often provide steep depth gradation and consequent vertical turnover of flora and faunal communities, with further spatial variability found in atoll systems where lagoons create relatively protected shallow habitats dominated by scleractinian reef-building coral, algae and seagrasses [26]. Understanding variability in predator fish assemblages across reef habitats in protected isolated reefs that are subject to low or no levels of human disturbance may provide further insights into the overall capacity of reefs to adapt to rapid ongoing change under ecologically optimal conditions [27].

Middleton Reef is a remote, protected (Marine National Park Zone, IUCN II) high-latitude atoll-like coral reef within the Lord Howe Seamount Chain. Diverse fish and predator dominated assemblages are an important characteristic of this reef system, which spans an enclosed inner lagoon to an outer steep shelf drop off [28,29]. Much of the knowledge of this system is limited to the shallow reef crest regions (typically <15 m) and lagoonal habitats consisting of seagrass, hard corals, macro- and turfing algae [30,31]. Here we extend this previous shallow water UVC sampling to 100 m depth using baited remote underwater stereo video (stereo-BRUV) to explore the potential for mesophotic habitats to act as refugia for key predatory fish species of this protected remote atoll. Specifically, we aimed to (i) quantify the relative abundance and body size structure of predator fish assemblages in the lagoon, sheltered north and exposed south aspects of Middleton Reef; (ii) understand the main habitat drivers associated with spatial variations in these predator assemblages, and (iii) determine if particular habitat variables are important in explaining the abundance of key species of the reef.

## 2 Materials and methods

### 2.1 Ethics statement

All fish in the current study were recorded with video using non-destructive techniques. Bait was used to attract fish following methods that were approved by the University of Tasmania Animal Ethics Committee (A0018195). Field work was completed under the Scientific Research Permit approved by Parks Australia (PA2019-00120).

### 2.2 Study location

Middleton Reef (29.4722˚ S, 159.1194˚ E) is located in the south-west Pacific and is approximately 200 km north of Lord Howe Island and 550 km east of mainland Australia (**Fig 1**). The reef is within a Marine National Park Zone (IUCN II) of the Lord Howe Marine Park that forms part of the Temperate East Australian Marine Park Network. At a juxtaposition of oceanographic influences, the region experiences warmer currents from the East Australian Current (EAC) and the cooler nutrient-rich Tasman Front [32]. The EAC varies in magnitude and frequency year to year [32]. Predominant seasonal weather and swell patterns originate from southeast trade-winds [33]. As found on other seamount features across eastern Australia, the southern aspect of the reef is exposed to prevailing conditions, comparative to the more leeward north of the reef [34].

Volcanic in origin, the planar surface of Middleton Reef rises abruptly from depths of ~ 2500 m and is shaped by wave erosion during subsequent seamount subduction [35]. The kidney shaped outer reef crest of Middleton Reef was established 5000 years ago, forming on top of an earlier Pleistocene carbonate framework [36]. Middleton Reef persists close to the typical 30˚ high latitudinal limit for shallow-water coral reef accretion [37]. High energy spur and groove features to the south of Middleton Reef reflect the prevailing conditions [38].

Habitats and faunal communities of Middleton Reef display features of both tropical coral reefs and cooler temperate systems [39]. Further, seasonal environmental variability defines periods of shallow-water coral growth, punctuated by phases of rubble accretion influencing reef structure [30]. The almost continuous perimeter reef crest is punctuated in the leeward north-eastern aspect, forming a relatively protected back reef and providing connectivity to the shallow sand dominated lagoon [38]. Sediments of the shallow inner lagoon are composed of gravels and finer sands made up of coral and coralline algae [36,40]. The shallow water benthic community is predominantly composed of a combination of algal turf, hard corals such as branching *Acropora* spp, macroalgae and crustose coralline algae [31]. Filter feeding organisms such as octo- and soft corals and sponges are supported on these outer slopes on similar geomorphic features in the region [41].

### 2.3 Sampling design

Stereo-BRUV deployments were designed to achieve spatially balanced sampling across depths from just below the surface to 100 m deep with a minimum spacing of 250m between concurrent deployments. The allocation of sample locations was done in R-package MBHdesign [42]. Inclusion probabilities within the sampling design were weighted towards complex and reef associated habitats from existing bathymetry data sourced from Geoscience Australia. Due to uncertainty in access to some regions of Middleton reef, a master sample of 300 sampling sites was initially generated. A master sample is essentially an ordered list of all possible sites in the study area (see Larsen et al. [43] for a detailed discussion on the use of master samples). If a sample in this list cannot be accessed, then the next site in the order is completed to maintain the initial spatial balance. However, some sample sites (particularly in the south and south-

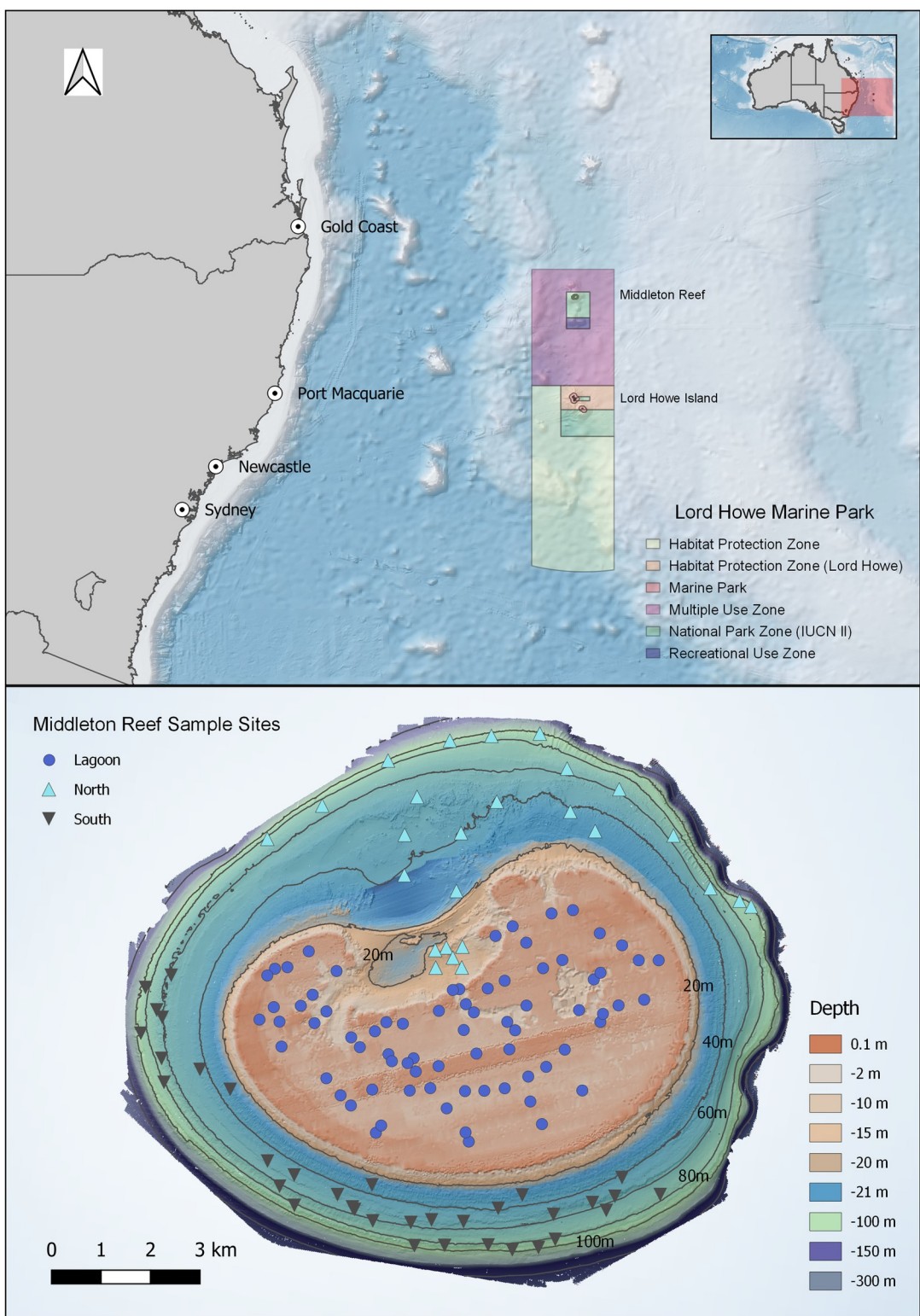

**Fig 1.** Location of Middleton Reef (a) within Lord Howe Marine Park, (b) sites sampled using stereo-BRUV across the sheltered north, exposed south and lagoon. The mainland Australia layer was retrieved from the geoBoundaries database (https://www.geoboundaries.org/data/1_3_3/zip/shapefile/AUS/). Marine Park boundary layer was extracted from the Collaborative Australian Protected Areas Database (http://www.environment.gov.au/fed/catalog/search/resource/details.page?uuid=%7BAF4EE98E-7F09-4172-B95E-067AB8FA10FC%7D). The underlying broad scale bathymetry is from GEBCO

Compilation Group (2021) Grid (doi:10.5285/c6612cbe-50b3-0cff-e053-6c86abc09f8f; https://www.gebco.net/data_and_products/gridded_bathymetry_data/). The finescale bathymetry was collected by authors and is also freely available at https://ecat.ga.gov.au/geonetwork/srv/eng/catalog.search#/metadata/144415. All are used under Creative Commons Attribution licence.

east) could not be completed due to large swells and associated navigational risks imposed by shallow reef crests. This resulted in under sampling of the shallows (i.e. 10-30m depths) along the southern margin of the reef.

## 2.4 Fish identification and image annotation

Overall, 131 stereo-BRUVs were deployed at Middleton Reef; 25 around the northern perimeter, 35 around the southern perimeter and 71 within the lagoon to sample fish assemblages and associated benthic habitats. Sampling was done during Austral summer 31st Jan–3rd Feb 2020 during daylight hours (0730–1700 hrs) following protocols by Langlois et al. [44]. For each stereo-BRUV, two cameras contained within waterproof PVC housings were fitted to a weighted metal frame with a soak time of 60 mins. GoPro6 cameras were used to sample the lagoon and Canon HFG and HFM models were used to sample outside the lagoon. A bait bag containing approximately 1kg of crushed pilchards (*Sardinops sagax*) was positioned 1.2 m from the camera lens. Baited video increases species richness and frequency sampled compared to unbaited video [45], is non-extractive, depth-independent [46] and suitable for sampling apex predators and mesopredators [47].

Each stereo-BRUV was calibrated before deployment and prior to image annotation using software CAL by SeaGIS [48]. A diode and clapper board were presented in the BRUV camera field of view commencing each deployment to allow for synchronisation of recordings. Stereo videos from Canon cameras were converted to.avi format with software Xilisoft [49], and GoPro.MP4 footage was imported directly into annotation software. Video annotation was conducted in EventMeasure Software by SeaGIS [48] (version 5.43 (64 bit)). All fish within an estimated 8 m of the stereo BRUV were annotated to the lowest taxonomic classification possible. The maximum number of individuals per species occurring within a single video frame was recorded (MaxN, hereafter referred to as relative abundance), and length measurements taken for each individual encountered [50]. Fork length was used to measure teleost fishes, from the most anterior tip of snout to middle of the caudal fin. Rays were measured across the widest portion of the disc. Total length was measured in sharks, from snout to posterior tip on caudal fin. For large schools of single species where not all individuals could be measured, at least 20 individual fish lengths were measured and multiplied for the remainder of the school. The lengths of individual fish were converted to weight using relationships obtained from FishBase [51]. Where species-specific relationships were not available, the relationship of a similar congener was used. Biomass estimates were calculated from the MaxN of each species in each deployment. Quality control and assurance of annotated outputs was carried out in CheckEM [52]. As sampling effort was different across each of the areas sampled, average abundance (based on summed MaxN per deployment) and average biomass (summed biomass per deployment) were standardised by dividing abundance or biomass by number of deployments per area. Predators were selected from the pooled fish assemblage based on trophic feeding groups from Reef Life Survey categorisation [53] and FishBase [51]. Efforts were made to reduce double counting of highly mobile predators (such as tiger shark, *Galeocerdo cuvier*) based on distinct markings and lengths of individuals. Where double counting was suspected the individual was excluded from the dataset, which only occurred in the large tiger shark individuals recorded in the lagoon.

## 2.5 Habitat classification

Habitat stills from stereo-BRUV were annotated in TransectMeasure software by SeaGIS [48] (version 3.31) to identify dominant benthic habitats and structural complexity following the CATAMI classification scheme for describing benthic habitats [54]. Following Langlois et al. [44], twenty points, arranged in a 4 x 5 grid pattern, were scored for relief and broad benthic habitat per deployment. Relief was scored as flat, low (<1m), moderate (1 – 3m), high (>3m). Benthic categories were classified as hard corals, octocoral / soft coral, sponges, invertebrate complex, rhodoliths, macroalgae, seagrass, consolidated (cobbles, boulders, rock) and unconsolidated (sand, pebbles, gravel, rubble) sediments.

## 2.6 Statistical analysis

Multivariate statistical analysis exploring the abundance and biomass structures of the predator fish assemblage was visualised through non-metric multidimensional scaling (nMDS) using a Bray-Curtis resemblance matrix in PRIMER [55]. No transformation was required for both datasets following visual interpretation of Shepard plots. A multivariate analysis of covariance (PERMANCOVA), and associated pairwise comparisons, was applied (Type III sum of squares, under a reduced model with 9999 permutations) accounting for the relationships between covariates (depth as a continuous variable and seabed habitat categories) on the abundance and biomass structures of the predator fish assemblage across area (north, south and lagoon) as a fixed factor. The SIMPER function was used to identify species contributing to observed percentage dissimilarity between areas around Middleton Reef based on abundance and biomass datasets. The relationship between predator fish assemblage structure (based on biomass or abundance) and seabed habitat variables and depth was analysed with distance-based linear model (DistLM) and visualised with distance-based redundancy analysis (dbRDA). Habitat variables and depth were log+1 transformed before being normalised. Vectors representing the most influential species/habitat variables were overlaid on dbRDA plots based on Pearson's correlations (>0.2 for species).

Several different analyses were conducted to understand the relationships between individual predatory fish and measured habitat variables. Species influential in distinguishing communities among areas around Middleton Reef (north, south, lagoon) from the SIMPER analysis were then used to test the relationships between habitat variables. Only species with sufficient abundance (MaxN > 15) were included. Generalized additive models (GAMs) with a full subset model selection (FSSgam) were used to test the relative importance of environmental characteristics on these select species, allowing for simultaneous evaluation of reef area, depth and habitat, in explaining variance in abundance, biomass and length distributions [56]. Full subset in GAM tests the relative importance of species metrics and for each environmental variable [56]. A Tweedie distribution was used for GAMs due to zero-inflation of abundance data [56]. Length and biomass were modelled using a Gaussian distribution [56]. Within the GAMs, *k* was limited to three degrees of freedom and model sizes limited to three terms to reduce overfitting of models. Model selection was based on Akaike's Information Criterion [57], optimised for small sample sizes (AICc; [58]). The relative importance of environmental variables was displayed in heat plots. Model performance was assessed using $R^2$.

Statistical analysis was carried out using PRIMER 6 (version 6.1.18) with the PERMANOVA (Permutational Multivariate Analysis of Variance) add-on (version 1.0.8) [55] and R version 3.6.0 [59]. Data manipulation was undertaken with R package tidyverse [60], and plots were created in R package ggplot2 [61]. Species accumulation curves for predator fish assemblages were generated in R package vegan (version 2.5–6) [62]. Curves were generated using

the "random" method with 9999 permutations to calculate accumulation of predatory fishes in the lagoon, northern and southern areas of the Middleton Reef across sampling effort.

## 3 Results

The abundance and biomass of predatory fishes (i.e. apex and mesopredators combined) contributed varying amounts to the trophic structure of fish assemblages at Middleton Reef (**Fig 2**). General patterns of increasing proportion of predator fish abundance (ranging from 9% in the lagoon to 68% at 91-120m in the south) and proportion of predator fish biomass (ranging from 51% in the <30m in the south to 93% at 61-90m in the north) with increasing depth was observed (**Fig 2**).

A total of 1044 individual predatory fish from 13 families and 36 species were recorded at Middleton Reef from 131 stereo-BRUV deployments across depths 0.3 to ~100 m (**Tables 1 and** S1). Abundance of predator fish ranged from 1 to 11 per deployment, with an average richness of 4.02 ± 2.18 S.D. and average abundance of 7.95 ± 27.96 S.D. across all deployments.

The species accumulation curve for the predator fishes in the lagoon reached asymptote, indicating that sampling effort adequately captured diversity (**Fig 3**). However, the species accumulation curve for the predator fishes in the north and south areas suggest that although outside reef areas were not as well sampled, there was a higher predator richness outside of Middleton Reef lagoon (**Fig 3**). Importantly, both north and south areas exhibited similar accumulation patterns suggesting that, although there were issues accessing the shallow reef margins (i.e. 10–20 m) in the south, it is unlikely to overly impact results.

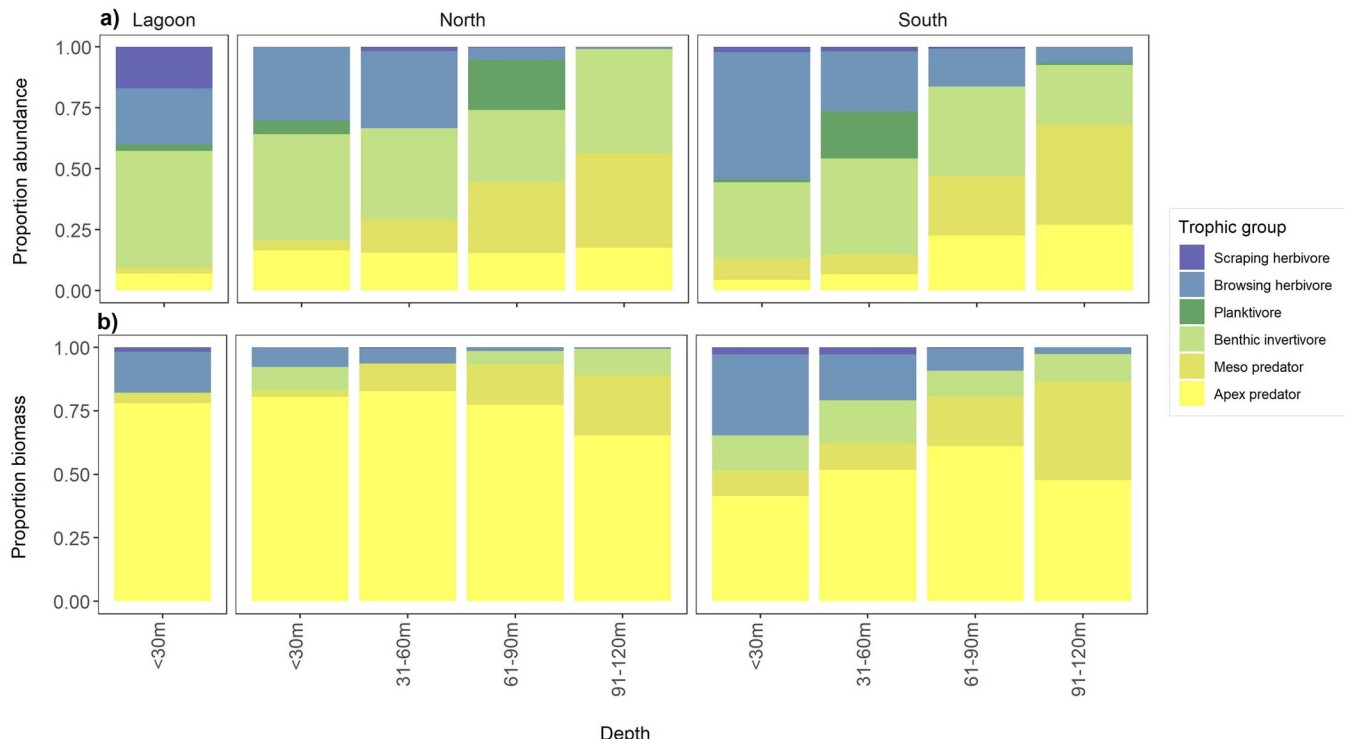

**Fig 2.** Comparison of changes in relative contribution (proportion of total fish assemblage) of trophic groups with depth using (**a**) abundance and (**b**) biomass data, respectively. Percentage abundance and biomass for each trophic group was calculated by summing all fish on all stereo-BRUV samples for all sites in a depth band.

**Table 1. Summary of predator fishes recorded in stereo-BRUV deployments on Middleton Reef.**

| | Reef area | | | |
| --- | --- | --- | --- | --- |
| | **Lagoon** | **North** | **South** | **Total** |
| Successful deployments | 71 | 25 | 35 | 131 |
| Depth range (m) | 1.0–10.4 | 11.2–98.8 | 28.6–95.2 | 1–98.8 |
| Total abundance (MaxN) | 357 | 303 | 384 | 1044 |
| Average abundance per deployment (mean ± SD) | 5.03 ± 37.21 | 12.12 ± 24.13 | 10.97 ± 22.14 | 7.97 ± 27.96 |
| Total biomass (tonne) sampled | 7.24 | 6.29 | 4.28 | 17.8 |
| Average biomass (tonne) per deployment (mean ± SD) | 0.10 ± 1.05 | 0.25 ± 0.67 | 0.12 ± 0.40 | 0.14 ± 0.74 |
| Predator richness | 24 | 25 | 25 | 36 |
| Average predator richness per deployment (mean ± SD) | 2.81 ± 1.33 | 4.91 ± 2.22 | 6.03 ± 1.87 | 4.02 ± 2.18 |
| Predator family richness | 10 | 12 | 9 | 15 |

### 3.1 Spatial patterns in abundance and biomass of predatory fishes

Predatory fishes on Middleton Reef were largely from five families: Carcharhinidae (344 individuals), Carangidae (278 individuals), Lutjanidae (181 individuals), Serranidae (128 individuals), Lethrinidae (29 individuals) (**Fig 4**). The most speciose predator families being Serranidae (11), Muraenidae (4) and Carangidae (4) (**S1 Table**). Commonly occurring predatory fishes included large-bodied apex and mesopredators, vulnerable and commercially valuable species (**S1 Table**). This included apex predators Galapagos shark (*Carcharhinus galapagensis*), tiger shark, yellowtail kingfish (*Seriola lalandi*) and black rockcod (*Epinephelus daemelii*). Rosy jobfish (*Pristipomoides filamentosus*), highfin amberjack (*Seriola rivoliana*) and red bass (*Lutjanus bohar*) were commonly occurring mesopredators (**S1 Table**; **S1 Fig**). Across all deployments, five species contributed to 90% of the total recorded predator biomass on the reef, being Galapagos shark (~54%), tiger shark (~17%), yellowtail kingfish (~12%), black rockcod (~4%) and rosy jobfish (~3%) (**S1 Table**; **S2 Fig**).

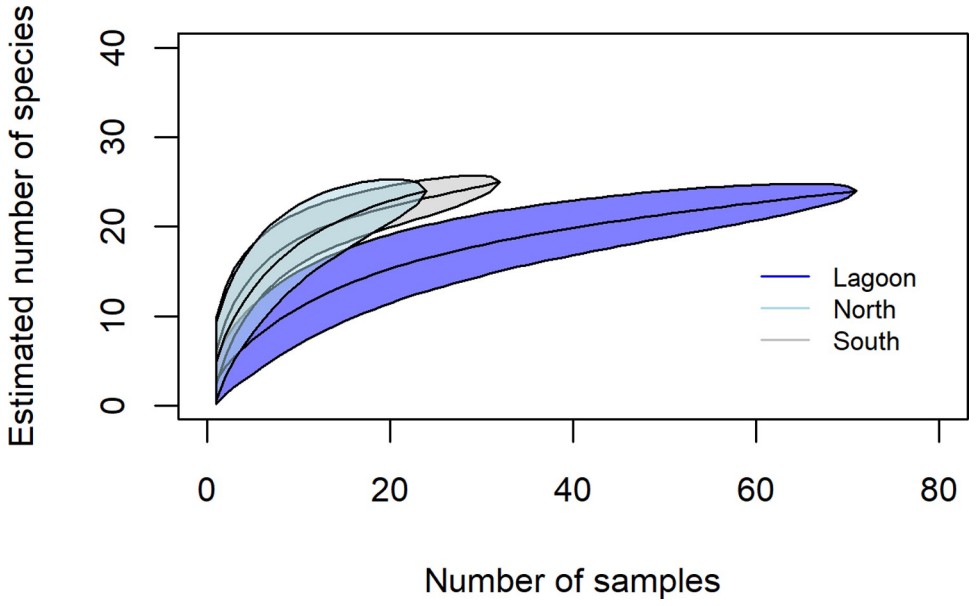

**Fig 3. Predator fish species accumulation curve showing sampling effort and estimated number of predator fishes sampled within lagoon, northern, and southern areas at Middleton Reef.**

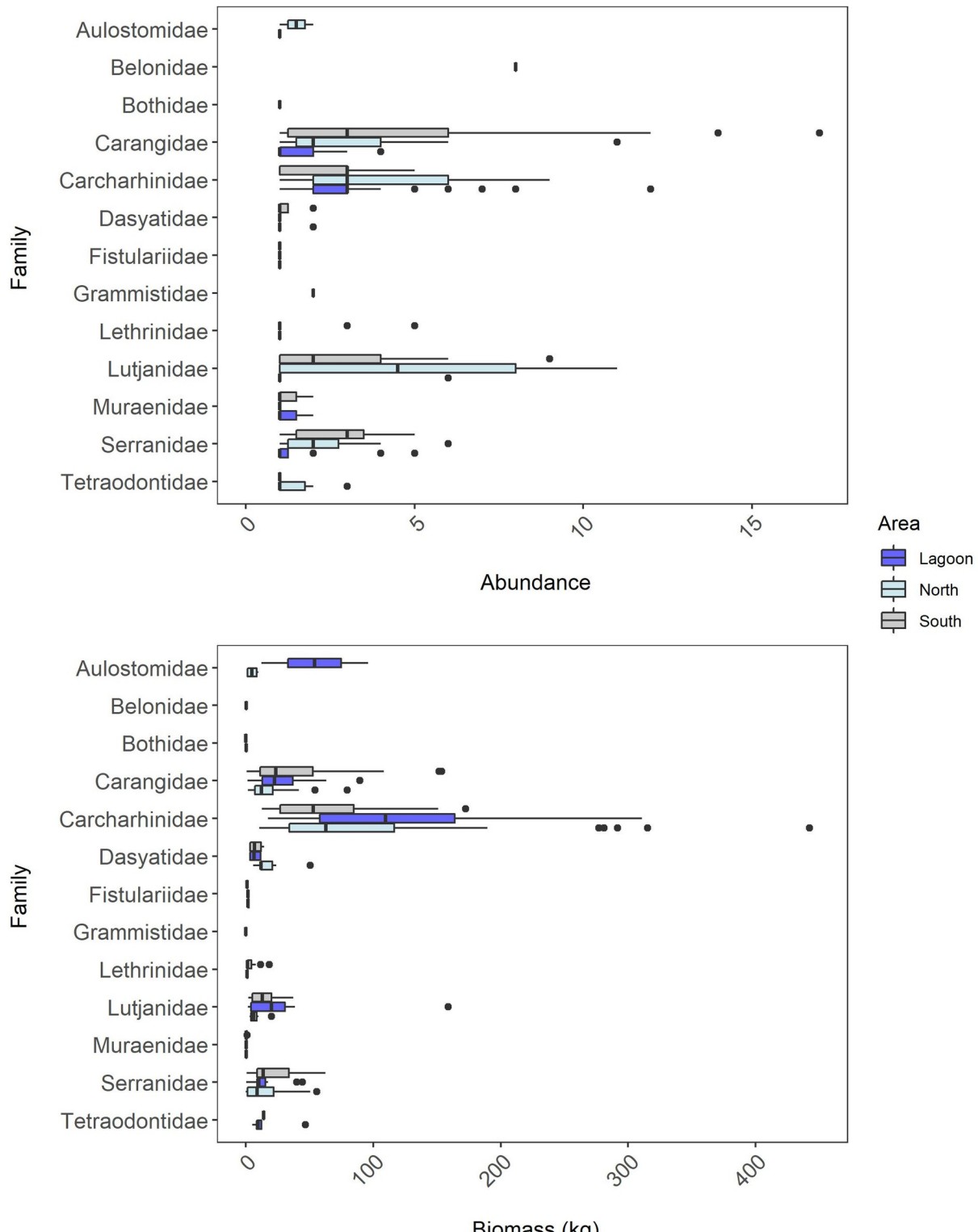

**Fig 4. Average abundance (MaxN) and biomass of predator families sampled during stereo-BRUV deployments within the lagoon, north and south of Middleton Reef.** Three outliers were removed from this plot to improve clarity: A MaxN of 33 individuals for Lutjanidae and individuals of 1000 kg and 1147 kg for Carcharhinidae.

The average abundance of predatory fishes per deployment was highest in the north of the reef, followed by the south, with both being greater than twice the average abundance of predatory fishes recorded in the lagoon (Table 1). The biomass of predators across the areas differed from abundance patterns, with the north area having two-fold higher biomass per deployment than lagoon and south areas, which are largely driven by the biomass of sharks (Table 1, Fig 4). Predatory fishes utilising the shallow inner lagoon habitat and the more protected northern areas of the reef, differed from those sampled on the more exposed southern areas (Table 1, Fig 5). There was a similar species richness of predatory fishes sampled in each area (Table 1). Six predator fish species were recorded in the inner lagoon (i.e. crocodile longtom (*Tylosurus crocodilus*), leopard flounder (*Bothidae pantherinus*), whitemouth moray (*Gymnothorax meleagris*), grey moray (*G. nubilus*), greyface moray (*G. thyrsoideus*), Pacific rockcod (*Trachypoma macracanthus*)). Three predator fish species were recorded only in the north of the reef (i.e. sandbar shark (*Carcharhinus plumbeus*), goldribbon cod (*Aulacocephalus temminckii*), flounder (*Bothus spp*)) and four predator fish species were recorded only in the south of the reef (i.e. black trevally (*Caranx lugubris*), redthroat emperor (*Lethrinus miniatus*), coral rockcod (*Cephalopholis miniata*), greasy rockcod (*Epinephelus tauvina*)).

Examination of the abundance and biomass structure of predator fish assemblages by nMDS indicated distinct partitioning between those of the protected shallow lagoon and those from outer areas sampled to the north and south (Fig 5). Analysis by PERMANCOVA indicated that the composition of predator fish abundance structure differed significantly between areas when accounting for significant effects of depth and seagrass cover (Table 2). Abundance assemblage structures differed significantly between the lagoon and north (*p*<0.0311), lagoon and south (*p*<0.0001), and north and south (*p*<0.0119) areas, indicated in pairwise comparisons.

The predator fish biomass assemblage structure across Middleton Reef also differed significantly by area accounting for the significant effects of depth (Table 2). Here, pairwise comparisons indicated that statistically significant differences in biomass assemblage structure were apparent between lagoon and south (*p*<0.0228), and between north and south areas (*p*<0.0163).

The SIMPER analyses for abundance of predator fishes showed that all areas had moderately high differences, with the lagoon and south areas of Middleton reef exhibiting the greatest dissimilarity (72.76%; Table 3). The difference between reef areas was characterised by changes in the abundance of 16 predator fishes, primarily Galapagos shark, yellowtail kingfish, highfin amberjack, rosy jobfish and to a lesser extent a range of smaller bodied mesopredators (Table 3, S1 Fig).

The cut-off point for low contributions was set at 90%.

By contrast, the SIMPER analyses for biomass of predator fishes had lower dissimilarity, with the north and south areas exhibiting the greatest difference (67.15%; Table 3). Similar to abundance, the biomass of Galapagos shark, yellowtail kingfish and rosy jobfish contributed most to the differences between reef areas (Table 3, S2 Fig).

## 3.2 Correlations of predatory fishes with environmental factors

The BEST procedure within the DistLM showed that 24.2% of the variation in the predator fish assemblage abundance structure could be explained by depth and cover of hard coral and bare sand (Table 4). The first two dbRDA axes explained 69.7% of the fitted variation (Fig 6A). Raw Pearson's correlations of each environmental factor with each dbRDA axis showed depth ($\rho$ = -0.98) correlated closely with the first dbRDA axis, with cover of octo/soft corals also partially correlated ($\rho$ = 0.55). The second axis was primarily associated with the cover of

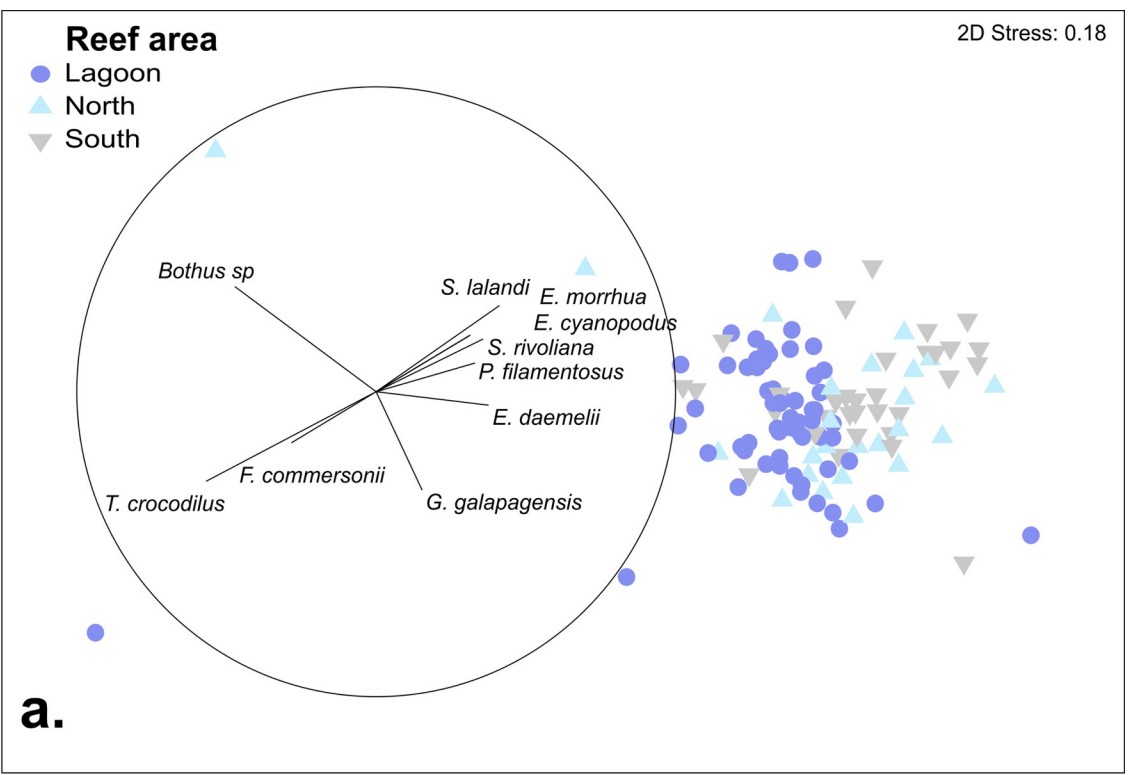

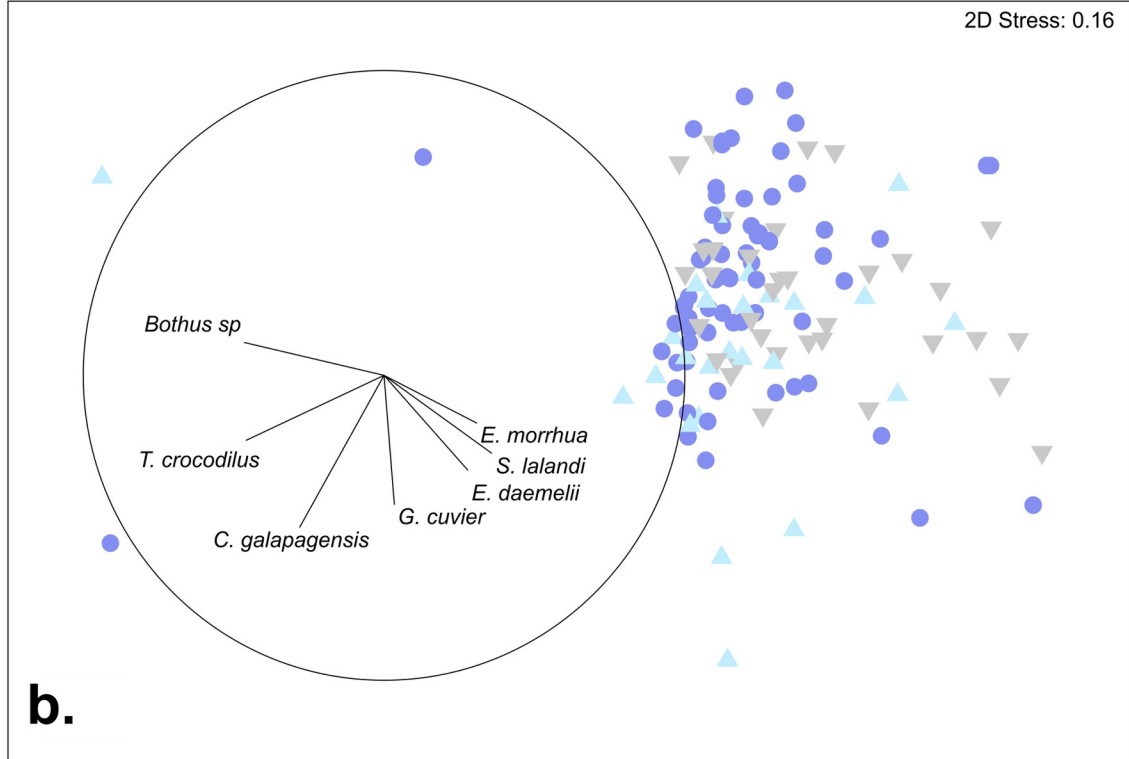

**Fig 5.** Non-metric multidimensional scaling (nMDS) ordination run with 25 random starts, a minimum stress = 0.01 with a Kruskal fit scheme, (a) predator fish abundance and (b) predator fish biomass. Species vectors (Pearson's correlation >0.2) are shown with the length of vectors representing measure of effect. Species vectors - *Bothus spp*, *T. crocodilus*, smooth flutemouth (*Fistularia commersonii*), *S. lalandi*, comet grouper (*Epinephelus morrhua*), *E. cyanopodus*, *S. rivoliana P. filamentosus*, *E. daemelii*, *C. galapagensis*, *G. cuvier*.

**Table 2. Results of PERMANCOVA assessing area differences in predator fish abundance and biomass assemblage structures accounting for variations in depth and habitat.**

| Variable | df | MS | Pseudo-F | p |
|---|---|---|---|---|
| **Abundance** | | | | |
| Rhodolith | 1 | 2503.1 | 1.3053 | 0.2327 |
| Rubble | 1 | 2645.1 | 1.3794 | 0.196 |
| Sand | 1 | 2927.8 | 1.5268 | 0.1619 |
| Gravel | 1 | 2762.1 | 1.4404 | 0.1841 |
| **Seagrass** | **1** | **4317.2** | **2.2514** | **0.0266** |
| Macroalgae | 1 | 2396.5 | 1.2497 | 0.2518 |
| Hard coral | 1 | 1026.7 | 0.53543 | 0.8151 |
| Octo/Soft coral | 1 | 2047.9 | 1.679 | 0.3746 |
| Sponges | 1 | 1159.7 | 0.60474 | 0.7662 |
| Turf | 1 | 1719 | 0.89644 | 0.485 |
| **Depth** | **1** | **20367** | **10.621** | **0.0001** |
| **Area** | **2** | **6735.7** | **3.512** | **0.0002** |
| Res | 112 | 1.1917 | | |
| Total | 125 | | | |
| **Biomass** | | | | |
| Rhodolith | 1 | 3377.3 | 1.6499 | 0.1368 |
| Rubble | 1 | 3709.9 | 1.8124 | 0.1053 |
| Sand | 1 | 3513.3 | 1.7163 | 0.121 |
| Gravel | 1 | 3500.9 | 1.7103 | 0.1263 |
| Seagrass | 1 | 2008.4 | 0.98118 | 0.4444 |
| Macroalgae | 1 | 3265.4 | 1.5953 | 0.1517 |
| Hard coral | 1 | 1712.5 | 0.83661 | 0.5226 |
| Octo/Soft coral | 1 | 3569.6 | 1.7439 | 0.1126 |
| Sponges | 1 | 1680.8 | 0.82112 | 0.545 |
| Turf | 1 | 2331.7 | 1.1391 | 0.3187 |
| **Depth** | **1** | **11626** | **5.6798** | **0.0001** |
| **Area** | **2** | **4687.9** | **2.2902** | **0.0082** |
| Res | 116 | 2046.9 | | |
| Total | 129 | | | |

Significant values are bold.

hard coral (ρ = 0.69). Raw Pearson's correlations with predator fish abundance with each of the dbRDA axes showed that a combination of yellowtail kingfish (ρ = -0.35), highfin amberjack (ρ = -0.41), spotcheek emperor (*Lethrinus rubrioperculatus;* ρ = -0.40), rosy jobfish (ρ = -0.40) and purple rockcod (ρ = -0.44) were most correlated with the first axis. The second axis was primarily associated with Galapagos shark (ρ = 0.43).

Similar patterns were evident in the predator assemblage biomass structure, with 16.8% of variation explained by depth and cover of hard coral (**Table 4**). The first two dbRDA axes explained 64.9% of fitted variation (**Fig 6B**). Raw Pearson's correlations of each environmental factor with these dbRDA axis showed depth (ρ = -0.95) was again correlated closely with the first dbRDA axis, with cover of octo/soft corals also partially correlated (ρ = -0.51). The second axis was primarily associated with the cover of hard coral (ρ = 0.52). Raw Pearson's correlations with predator biomass with each of the dbRDA axes showed that a combination of highfin amberjack (ρ = -0.44), spotcheek emperor (ρ = -0.34), rosy jobfish (ρ = -0.38), purple

**Table 3. The SIMPER analysis using Bray-Curtis similarity index identifying key predator fish abundance and biomass contributions to dissimilarities between areas sampled at Middleton Reef.**

| Scientific name | Abundance Dissimilarity (%) | | |
|---|---|---|---|
| | Lagoon–North | Lagoon–South | North–South |
| Average total | 72.22 | 72.76 | 70.40 |
| Carcharhinus galapagensis | 17.31 | 11.62 | 11.58 |
| Seriola lalandi | 8.62 | 12.34 | 10.44 |
| Seriola rivoliana | 0 | 7.22 | 5.46 |
| Pristipomoides filamentosus | 15.07 | 6.7 | 14.08 |
| Epinephelus daemelii | 5.22 | 3.53 | 2.67 |
| Epinephelus maculatus | 2.29 | 1.44 | 2.05 |
| Epinephelus cyanopodus | 1.47 | 2.27 | 1.92 |
| Epinephelus morrhua | 0.00 | 0.00 | 1.05 |
| Epinephelus rivulatus | 1.03 | 2.2 | 1.94 |
| Carangoides orthogrammus | 1.62 | 2.24 | 0 |
| Lethrinus rubrioperculatus | 0.00 | 4.04 | 2.74 |
| Lethrinus miniatus | 0 | 1.71 | 0.00 |
| Aprion virescens | 1.59 | 2.27 | 1.87 |
| Lutjanus bohar | 1.31 | 3.13 | 2.19 |
| Galeocerdo cuvier | 1.97 | 1.06 | 1.14 |
| Variola louti | 0 | 1.11 | 0.00 |
| **Species** | **Biomass Dissimilarity (%)** | | |
| | Lagoon–North | Lagoon–South | North–South |
| Average Total | 66.44 | 65.42 | 67.15 |
| Carcharhinus galapagensis | 30.11 | 26.84 | 25.82 |
| Seriola lalandi | 6.34 | 11.91 | 8.8 |
| Seriola rivoliana | 0 | 3.38 | 2.59 |
| Pristipomoides filamentosus | 4.94 | 3.55 | 5.48 |
| Epinephelus daemelii | 2.22 | 4.68 | 3.22 |
| Epinephelus cyanopodus | 0 | 2.06 | 1.61 |
| Galeocerdo cuvier | 13.41 | 5.04 | 11.41 |
| Aulostomus chinensis | 1.88 | 0 | 1.62 |

rockcod ($\rho$ = -0.38) and comet grouper ($\rho$ = -0.32) were most correlated with the first axis. The second axis was primarily associated with Galapagos shark ($\rho$ = 0.37).

### 3.3 Habitat associations for key predatory fishes

The most parsimonious models of species-habitat relationships based on GAMs varied in overall prediction accuracy between species, with best models ranging from a low $R^2$ of 0.06

**Table 4. Distance based linear model (DistLM) showing environmental factors identified using the BEST procedure significantly correlated with assemblage structure of predator abundance and biomass on Middleton Reef.**

| Environmental factors | Pseudo-F | p | Prop. variation |
|---|---|---|---|
| **Abundance** | | | |
| Depth | 17.99 | 0.0001 | 0.1826 |
| Hard coral | 3.37 | 0.0005 | 0.2284 |
| Sand | 2.4 | 0.0138 | 0.2421 |
| **Biomass** | | | |
| Depth | 8.52 | 0.0001 | 0.1423 |
| Hard coral | 2.38 | 0.0252 | 0.1681 |

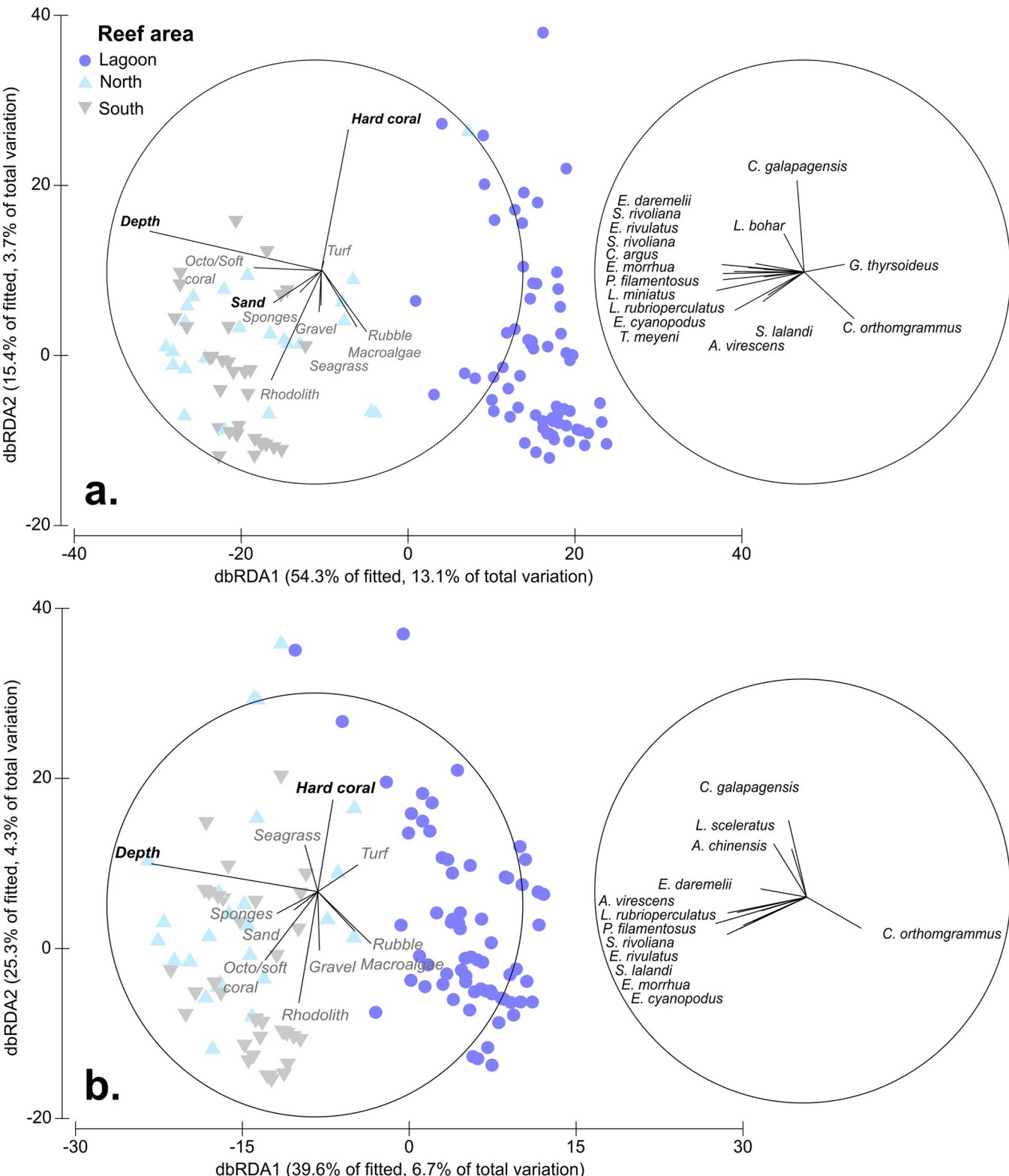

**Fig 6.** Distance-based redundancy analysis (dbRDA) plot of Bray-Curtis dissimilarities showing the relationship between predator fish assemblage structure and environmental factors at Middleton Reef based on **(a)** abundance **(b)** biomass. Length of vectors display the strength of variables' influence. Bold environmental vectors are those selected using the BEST procedure in DistLM. Species vectors are those with >0.2 Pearson's correlation with the first two dbRDA axes.

**Table 5. Top generalized additive models (GAMs) for predicting the abundance distribution, biomass distribution and length distribution of species of interest from full subset analysis.**

|  | Scientific name | Model | AICc | ωAiCc | $R^2$ | eDF |
|---|---|---|---|---|---|---|
| Abundance | *Carcharhinus galapagensis* | Seagrass + hard coral + turf | 424.50 | 0.05 | 0.12 | 4.95 |
|  | *Seriola lalandi* | Rhodolith + seagrass + sponges | 242.37 | 0.49 | 0.47 | 4.90 |
|  | *Seriola rivoliana* | Depth + octo/soft coral | 99.20 | 0.12 | 0.25 | 3.91 |
|  | *Epinephelus daemelii* | Rhodolith + sponges + turf | 45.65 | 0.34 | 0.57 | 4.26 |
|  | *Epinephelus rivulatus* | Octo/soft coral + turf | 2.31 | 1.00 | 0.98 | 4.90 |
|  | *Lethrinus rubrioperculatus* | Depth + turf | 50.57 | 0.79 | 0.96 | 3.99 |
|  | *Pristipomoides filamentosus* | Depth + gravel + sponges | 96.87 | 0.97 | 0.70 | 5.81 |
|  | *Aprion virescens* | Rhodolith + sponges | 11.77 | 0.26 | 3.00 | 0.09 |
|  | *Lutjanus bohar* | Rubble | 37.63 | 0.44 | 0.80 | 2.94 |
| Biomass | *Carcharhinus galapagensis* | Gravel + rubble + seagrass | 7945.89 | 0.14 | 0.06 | 4.83 |
|  | *Seriola lalandi* | Macroalgae + octo/soft coral + seagrass | 3947.40 | 0.96 | 0.25 | 6.28 |
|  | *Seriola rivoliana* | Octo/soft coral + seagrass + turf | 1359.57 | 0.11 | 0.18 | 4.68 |
|  | *Epinephelus daemelii* | Gravel + sand + turf | 911.09 | 0.08 | 0.28 | 5.74 |
|  | *Epinephelus rivulatus* | Area + sponges | 227.14 | 0.30 | 0.65 | 4.52 |
|  | *Lethrinus rubrioperculatus* | Octo/soft coral | 278.29 | 0.23 | 0.28 | 2.72 |
|  | *Pristipomoides filamentosus* | Area + rhodolith x area + turf x area | 2493.68 | 0.33 | 0.29 | 7.87 |
|  | *Aprion virescens* | Macroalgae + sand + sponges | 380.73 | 0.08 | 0.55 | 4.88 |
|  | *Lutjanus bohar* | Sand + turf | 427.23 | 0.22 | 0.52 | 4.02 |
| Length | *Carcharhinus galapagensis* | Rhodolith + rubble + seagrass | 3314.86 | 0.12 | 0.10 | 5.63 |
|  | *Seriola lalandi* | Macroalgae + octocoral + seagrass | 2148.27 | 0.59 | 0.26 | 5.46 |
|  | *Seriola rivoliana* | Octo/soft coral + seagrass + turf | 751.32 | 0.10 | 0.21 | 4.00 |
|  | *Epinephelus daemelii* | Sand + hard coral + turf | 493.71 | 0.40 | 0.55 | 5.68 |
|  | *Epinephelus rivulatus* | Area | 132.94 | 0.30 | 0.54 | 3.00 |
|  | *Lethrinus rubrioperculatus* | Depth + turf | 184.76 | 0.11 | 0.51 | 4.68 |
|  | *Pristipomoides filamentosus* | Rhodolith + sand + turf | 1228.08 | 0.20 | 0.23 | 5.72 |
|  | *Aprion virescens* | Depth + sand | 242.39 | 0.20 | 0.48 | 3.83 |
|  | *Lutjanus bohar* | Turf | 190.36 | 0.43 | 0.57 | 2.67 |

Reported metrics are the AICc, Akaike Information Criterion; •AICc, the Weighted Akaike Information Criterion; explained variance, $R^2$ and effective degrees of freedom, eDF.

for the biomass of Galapagos shark to a high of 0.98 for the biomass of Chinaman rockcod (*Epinephelus rivulatus*) (**Table 5**).

Neither specific habitat type, area or depth emerged as the important overall driver in explaining the abundance, biomass and lengths of many of the individual species modelled against environmental characteristics on Middleton Reef (**Fig 7**). The mobile predators Galapagos shark, yellowtail kingfish, highfin amberjack, green jobfish and red bass did not show particular affinities with habitat characteristics, although some patterns were evident for the reef associated species. Octo/soft coral and turf algae cover were the most important habitats in explaining abundance for the serranid, Chinaman rockcod. For mesophotic species spotcheek emperor and rosy snapper, depth, turf algae and sponges respectively were the most important variables in explaining abundance distributions.

## 4 Discussion

We found that the remote and protected Middleton Reef supports a highly diverse, abundant predator fish assemblage, with five species that contributed to 90% of the total recorded

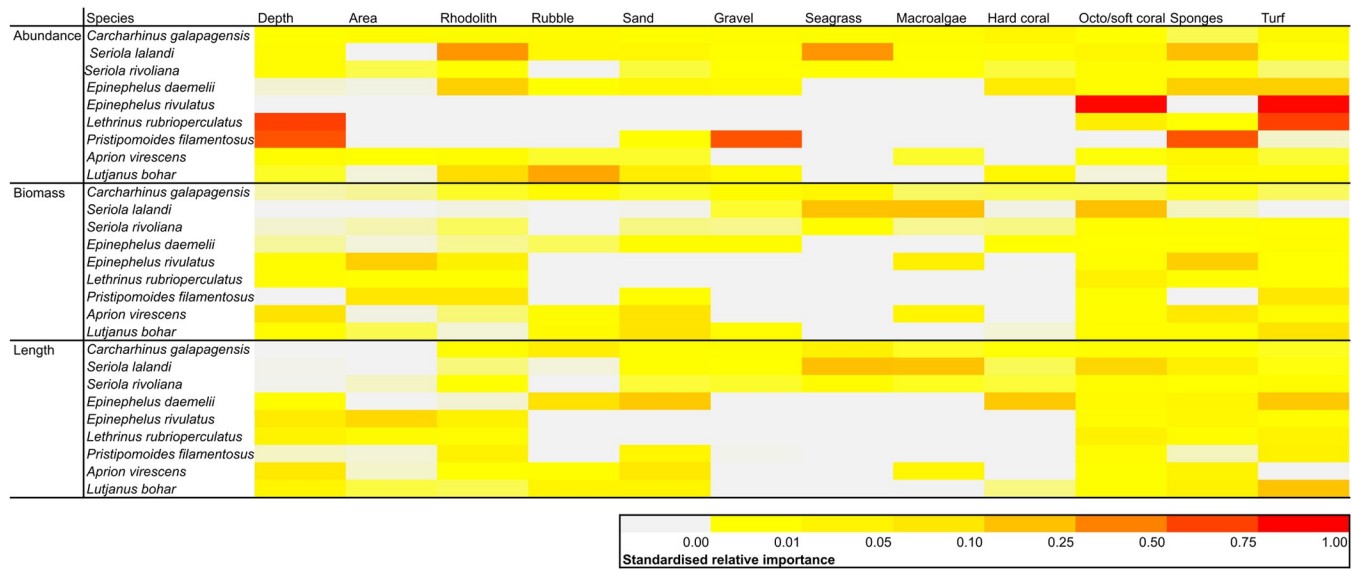

**Fig 7. Heatmaps displaying the standardised variable importance scores from full subset GAM analysis to predict the abundance distribution, biomass distribution and length distributions of selected predatory fishes on Middleton Reef.** Standardising was done against deviance explained for each model.

predator biomass on the reef, dominated by Galapagos shark (~54%), tiger shark (~17%), yellowtail kingfish (~12%), black rockcod (~4%) and rosy jobfish (~3%). Our findings show there were significant predator assemblage differences in richness, abundance and biomass evident across lagoonal to mesophotic shelf habitats. Differences in abundance and biomass assemblage structures were largely associated with representatives from Carcharhinidae and Carangidae families. Importantly, assemblages showed significant associations with seabed habitat (e.g. hard coral cover), area (i.e. those found in the north compared to the south) and depth.

Hard coral cover was a defining feature of shallow seabed areas at Middleton Reef, representing the tropical influence of this high-latitude system. It is well known that hard coral cover and diversity within tropical reef ecosystems influence the fishes utilising them [63], and can impact the assemblage structure of fishes across all trophic levels [64–66]. The hard coral growth forms at Middleton Reef are variable (dominated by a variety of submassive and digitate morphologies) and are known to support an array of potential prey in the form of smaller bodied reef associated fishes [31]. The south of Middleton Reef is characterised by a complex mesophotic environment [67] associated with filter feeding organisms and high energy spur and groove features [38]. Complex habitat structures support species with varied life history characteristics, such as those suitable for ambush style predation used by Serranidae species, which utilise caves, boulders and overhangs [68,69]. Highest relative abundances of Serranidae species in the south of Middleton Reef (e.g. Peacock rockcod (*Cephalopholis argus*), coral rockcod, purple rockcod, black rockcod, comet grouper and Chinaman rockcod) coincide with topographically complex habitats (e.g. spur and groove habitat features [69]).

Previous studies have also linked the distribution and density of top order predatory fishes to prey availability [70,71]. In fact, Boaden and Kingsford [72] suggest that high densities of predators are stronger predictors of prey density than the structure and composition of habitat. Given the predator assemblage at Middleton Reef was dominated by highly mobile Carangidae and Carcharhinidae, the association with structurally complex habitats (such as hard coral) is perhaps a reflection of habitat preferences of prey as opposed to predator-habitat relationships. One notable feature of the predator fish assemblage at Middleton Reef was

significant structuring by depth. As found in other remote locations, while overlap in distribution between shallow and mesophotic habitats occurred for some predators, clear depth preferences were evident for others [73]. For example, the highly abundant shark species, Galapagos shark, was broadly distributed from shallow to mid- and upper- mesophotic depths at Middleton Reef, a similar range to that recorded elsewhere [74]. By contrast, commercially important rosy jobfish was only recorded in the mid-mesophotic depths (i.e. ~70–90 m) at Middleton Reef. Interestingly this commercially valuable species has a broad geographic range, being widely distributed across mesophotic habitats within the Great Barrier Reef [13] and into the Indo-Pacific. Due to this wide geographic range, it is thought that these atoll-like seamounts (such as Middleton Reef) may provide connectivity across the region, acting as 'stepping stones' [75].

Previous studies have shown that larger individuals and higher abundances of fishery targeted species are often more abundant in deeper coral reefs [19]. This is because fishing effort is often focused nearer to shore where depths are shallower. Thus, the harder to access deeper regions potentially provide important refuge from fishing pressure [76]. Prior to protection, the historical fishing pressure at Middleton Reef was extremely low due to its remote location. Our work suggests that patterns in abundance and biomass of the mid-trophic level predators (e.g., spotcheek emperor, comet grouper) positively associated with depth may be more closely attributed to predator fishes in this region being naturally more abundant at mesophotic depths—potentially preying on the abundant small planktivorous species commonly aggregated near the steeper flanks of these atoll-like seamounts [77–79].

The lagoon habitats of Middleton Reef may provide a nursery ground for Galapagos shark as all individuals recorded in the study were immature. In addition, the entrances to the lagoon were frequented by Galapagos shark and tiger shark. Previous research suggests that sharks often aggregate at these locations during tide changes, with this behaviour linked to prey movement between inner and outer reef habitats [80]. Additionally, flushing of atoll environments through tidal movement increases phytoplankton productivity and mixing in adjacent waters [81], which may influence the abundance of prey items for sharks in this area. The north of Middleton Reef consists of a wider, sediment dominated shelf margin with concentric ridge-like reef features [67], and is likely to be more exposed to the warmer waters of the East Australian Current. The south of Middleton Reef is exposed to south-east trade winds and associated swells, potentially leading to upwelling events. Similar dynamics is evident at other oceanic reefs with, for example, reef fish assemblages in the North Pacific of Costa Rica being influenced not only by seasonal upwelling events, but also by the interplay with other seabed habitat features [82]. Hence, the differences in fine- and broad-scale geomorphological habitat features interacting with differing oceanographic conditions are potentially driving the predator fish assemblages at Middleton Reef.

## 5 Conclusions

Declines in predatory fishes are far reaching across the globe, including the Indo-Pacific, with remote features afforded some protection through their inaccessibility [22]. Therefore, understanding spatial distributions of species on these remote oceanic locations is key for monitoring change across time and building resilience against future threats [83]. In addition to its recent no-take status, the predator fish assemblages of Middleton Reef appear to also benefit from its geographically remote location. This was reflected by a high abundance and biomass of top order predators across shallow lagoon and mesophotic habitats. The no-take protection status of Middleton Reef appears well considered as it hosts high abundances of vulnerable species, such as the black rockcod. Additionally, the lagoon habitat represents a potential

nursery for the Galapagos shark in which the distribution and habitat use of the adult population remains unknown. In addition to apex predators, there is a broad diversity of pelagic and reef associated mesopredatory fishes residing on Middleton Reef, typically partitioned across depth and seabed habitats. This study forms a baseline for future monitoring of predatory fishes in the region and adds to the growing body of literature highlighting the need to ensure that lagoon, shallow and mesophotic habitats are adequately protected, as they support ecologically and economically important predator fish assemblages.

## Supporting information

**S1 Fig. Spatial distribution of abundance for predatory fishes recorded in stereo-BRUV deployments on Middleton Reef.** a. *Carcharhinus galapagensis*, b. *Galeocerdo cuvier*, c. *Epinephelus daemelii*, d. *E. cyanopodus*, e. *Lethrinus rubrioperculatus*, f. *Lutjanus bohar*, g. *Seriola lalandi*, h. *S. rivoliana*, i. *Pristipomoides filamentosus*, j. *Aprion virescens*.
(TIF)

**S2 Fig. Spatial distribution of biomass for predatory fishes recorded in stereo-BRUV deployments on Middleton Reef.** a. *C. galapagensis*, b. *G. cuvier*, c. *E. daemelii*, d. *E. cyanopodus*, e. *L. rubrioperculatus*, f. *L. bohar*, g. *S. lalandi*, h. *S. rivoliana*, i. *P. filamentosus*, j. *A. virescens*.
(TIF)

**S1 Table. Predatory fishes recorded in stereo-BRUV deployments on Middleton Reef.** Abundance based on summed MaxN; standardized abundance based on number of deployments per area (lagoon 71, north 25, south 35) and calculated total biomass.
(DOCX)

## Acknowledgments

This work was undertaken for the Marine Biodiversity Hub, a collaborative partnership supported through the Australian Government's National Environmental Science Program (NESP). NESP Marine Biodiversity Hub partners include the University of Tasmania, CSIRO, Geoscience Australia, Australian Institute of Marine Science, Museums Victoria, Charles Darwin University, the University of Western Australia, Integrated Marine Observing System, NSW Office of Environment and Heritage, NSW Department of Primary Industries. We also acknowledge the support provided by the Director of National Parks, and particularly Dr Cath Samson from Parks Australia. We thank the master and crew of TV Bluefin and support staff from the Australian Maritime College. Dr Paulus Justy W. Siwabessy is thanked for postprocessing the multibeam sonar dataset. Thanks also to Justin Hulls (UTas), Brett Louden and Matt Hammond (NSW DPI) for their technical assistance in the collection and annotation of stereo BRUV footage. Antonia Cooper, Dr Mike Cappo, Dr Dianne Bray, and Dr Martin Gomon are thanked for their assistance in the identification of fish species. KB would also like to thank Dr Christina Buelow for her insight. AC publishes with permission from the CEO of Geoscience Australia. Authors would like to thank Associate Professor Alan Jordan, the Editor and Reviewers for their comments that greatly improved the manuscript.

## Author Contributions

**Conceptualization:** Kristy Brown, Jacquomo Monk, Neville Barrett.

**Data curation:** Kristy Brown, Jacquomo Monk, Joel Williams.

**Formal analysis:** Kristy Brown, Jacquomo Monk.

**Funding acquisition:** Jacquomo Monk, Andrew Carroll, Neville Barrett.

**Methodology:** Jacquomo Monk, Andrew Carroll, David Harasti, Neville Barrett.

**Project administration:** Andrew Carroll, Neville Barrett.

**Resources:** Jacquomo Monk, David Harasti, Neville Barrett.

**Visualization:** Kristy Brown, Jacquomo Monk.

**Writing – original draft:** Kristy Brown, Jacquomo Monk.

**Writing – review & editing:** Kristy Brown, Jacquomo Monk, Joel Williams, Andrew Carroll, David Harasti, Neville Barrett.

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
