## [Decision Letter · Decision Letter 0]

26 Oct 2021

PONE-D-21-27786Depth and benthic habitat influence shallow and mesophotic predatory fishes on a remote, high-latitude coral reefPLOS ONE

Dear Dr. Monk,

Thank you for submitting your manuscript to PLOS ONE. After careful consideration, we feel that it has merit but does not fully meet PLOS ONE’s publication criteria as it currently stands. Therefore, we invite you to submit a revised version of the manuscript that addresses the points raised during the review process. Both reviewers believe that this manuscript is a valuable addition to the literature, and helps fill a gap in our knowledge about fish in mesophotic environments. However, they also both comment that the methods require some improvement, with a focus on clarifying the classification of fishes (e.g., between top-predator and mesopredator, and standardisation of names - particularly reviewer 2), more information on how depth zones were sampled including the gap in sampling shallow zones and how this may influence conclusions, potentially using zone rather than depth in the analysis in order to strengthen the argument concerning community changes across a depth gradient, and a little more polish generally within the methods section to bring it up to the standards of the rest of the manuscript. While all comments should be addressed in the response to reviewers, please pay particular attention to reviewer one's critique of the interpretation of the community patterns shown in Figure 2.

We look forward to receiving your revised manuscript.

Kind regards,

Fraser Andrew Januchowski-Hartley, Ph.D.

Academic Editor

PLOS ONE

Journal Requirements:

“This work was undertaken for the Marine Biodiversity Hub, a collaborative partnership supported through funding from the Australian Government’s National Environmental Science Program (NESP). NESP Marine Biodiversity Hub partners include the University of Tasmania, CSIRO, Geoscience Australia, Australian Institute of Marine Science, Museums Victoria, Charles Darwin University, the University of Western Australia, Integrated Marine Observing System, NSW Office of Environment and Heritage, NSW Department of Primary Industries.

We also acknowledge the support provided by the Director of National Parks, particularly Dr Cath Samson from Parks Australia. We thank the master and crew of TV Bluefin and support staff from the Australian Maritime College.”

4. We note that you have stated that you will provide repository information for your data at acceptance. Should your manuscript be accepted for publication, we will hold it until you provide the relevant accession numbers or DOIs necessary to access your data. If you wish to make changes to your Data Availability statement, please describe these changes in your cover letter and we will update your Data Availability statement to reflect the information you provide

5. We note that Figure 1 in your submission contain map images which may be copyrighted. All PLOS content is published under the Creative Commons Attribution License (CC BY 4.0), which means that the manuscript, images, and Supporting Information files will be freely available online, and any third party is permitted to access, download, copy, distribute, and use these materials in any way, even commercially, with proper attribution. For these reasons, we cannot publish previously copyrighted maps or satellite images created using proprietary data, such as Google software (Google Maps, Street View, and Earth). For more information, see our copyright guidelines: http://journals.plos.org/plosone/s/licenses-and-copyright.

    1. You may seek permission from the original copyright holder of Figure(s) [#] to publish the content specifically under the CC BY 4.0 license. 

Maps at the CIA (public domain): https://www.cia.gov/library/publications/the-world-factbook/index.html and https://www.cia.gov/library/publications/cia-maps- publications/index.html

Reviewers' comments:

Reviewer's Responses to Questions

**Comments to the Author**

1. Is the manuscript technically sound, and do the data support the conclusions?

Reviewer #1: Yes

Reviewer #2: Yes

2. Has the statistical analysis been performed appropriately and rigorously? 

Reviewer #1: Yes

Reviewer #2: Yes

3. Have the authors made all data underlying the findings in their manuscript fully available?

Reviewer #1: Yes

Reviewer #2: Yes

4. Is the manuscript presented in an intelligible fashion and written in standard English?

Reviewer #1: Yes

Reviewer #2: Yes

5. Review Comments to the Author

Reviewer #1: This is one of the few studies that documented patterns in the abundance and biomass of predatory fish assemblages in remote atoll seamount MPAs. The study used baited cameras to efficiently sample large areas of the lagoon and reef slope of the atoll, extending to the mid-mesophotic zone. The data presented are a valuable contribution to the coral reef ecological literature as previous studies of reef fish assemblages on remote atolls are mostly limited to euphotic depths. The ms is relatively well written but there is room for some improvement. I hope the general comments below will help improve the paper. Also listed below are minor comments, suggestions and corrections.

General comments

1. The work highlighted the important roles of predatory fishes in the trophic ecology of reefs, the concerns about their dwindling numbers due to overfishing, and the importance of establishing baseline reference points for their spatial distribution (L39-49). However, it was difficult to appreciate the data presented in light of these points without showing some data on the overall fish assemblage. What proportion of the species richness, abundance and biomass of the overall fish assemblage in the atoll can be accounted for by predatory fishes? How do these proportions change with location and depth within the atoll? Do these proportions differ with other remote atolls in other regions that are open or closed to fishing? Addressing these questions may enhance the value of the ms. This seems fairly easy to do because the patterns presented here were simply extracted from the pooled fish assemblage data (L151-152).

2. Distinguishing patterns in the top predators from those of mesopredators seems useful and important to this ms. In many sections (e.g. L39, L58, L353-358, L399-405), the text seems to suggest a desire to highlight patterns in the abundance and biomass of top predators like sharks, large carangids and large groupers. There was also a desire to highlight the mesopredators (L376, L406). However, the ms did not explicitly identify which ones are top predators or mesopredators, for instance in Tables 1, 3, 5, and S1.

3. The study highlighted significant differences in the structure of predatory fish assemblages between the upper- and mid-mesophotic zones and the lagoon (L333-334). However, the results did not really support this because partitioning into upper (30-60 m) and mid (60-90) mesophotic zones (L54) was not explicitly considered in the sampling design and data analysis (i.e. depth was treated as a continuous variable).

4. The depth range between about 11 to 29 m was not sampled in the south part of the atoll (Table 1). Was this a consequence of spatially balanced sampling that was weighted towards more complex habitat (L118)? Or was this simply a consequence of difficulties sampling that depth range in the south part of the atoll due to wave exposure? In any case, this gap in sampling should be clarified in the methods. More importantly, its potential implications for the conclusions must be discussed.

Minor comments, suggestions and corrections

L18-20 – Sentence about Middleton Reef seems out of place here. It can be moved further down, around L24 where the authors described where the study was conducted.

L28 – 90% of the total fish biomass or just the total biomass of large predators? Needs clarification.

L47 – Knowledge may be a better word to use here than understandings.

L81 – key predatory species?

L85 – Sentence needs improvement. Suggestion: “Bait was used to attract fish following methods that were approved by the Animal Ethics Committee…”

L104 – Specify that you are referring to habitats and faunal communities of Middleton Reef

L125 – austral (as opposed to boreal), not Astral

L126 – Odd to read numbers here instead of the authors

L131 – Must be camera lens, not field (and field of view is another matter)

L133 – higher order predators and mesopredators

L135 – were instead of was

L147 – Needs correction and further explanation. Parameters a and b of length-weight models for each species are what can be sourced from Fishbase, not biomass calculations. Were length-weight models (in FL) available for all species? If not (or only available for TL), what was the procedure?

L148 – spell out QA/QC (quality control and assurance?)

L151 – by dividing abundance AND biomass?

L154 – Suggest providing an idea of the number of times double-counting was suspected for these mobile predators, and the procedure taken to omit suspected repeats counts

L161 – of THE predatory fish assemblage…

L164-165 – Clarify that you are referring to abundance of the selected predatory fishes only.

L168 – Suggest modifying to “abundance and biomass structures of the predator assemblage across area (north, south and lagoon) as a fixed factor”

L200 – Check value for average abundance across all deployments stated in text against Table 1

L202-203 – Figure 2 does not convincingly show that sampling in the north and south areas adequately captured predator diversity. I tend to disagree that predatory species accumulation curves “largely reached asymptotes” in these regions. Instead, what Fig. 2 suggests is that the north and south areas have a higher species richness of predatory fish and that these areas were not as well-sampled as the lagoon. I think the original statement should be toned down and the higher species richness outside of the lagoon of Middleton Reef should be highlighted.

L223-226 – This sentence was a little confusing because it shifts its reference from total biomass to average biomass per deployment. Also, it should refer to Table 1 not S1.

L242, 246 – fish not fishes

L251-255 – Check sentence construction. This is a very long sentence that is hard to read.

L302 – GAMs not GAM’s

L304 – bohar not Bohar

L306-310 – Again, a very long sentence that is difficult to understand. Suggest reconstructing or splitting.

L330 – five species THAT contributed to…

L353 – remove semicolon

L354 – Odd to read a number here instead of the authors

L374 – Remove open parenthesis

L402-403 – E. daemelii is listed as near threatened (NT) which is NOT threatened, at least not yet.

L404-405 – I didn’t fully understand the latter part of this sentence. It seems to say that C. galapagensis are likely to be vulnerable even if adult distributions of the species are unknown. But the species is listed as least concern (LC) – Table S1

Fig. 3 caption – Were the outliers removed from this plot all records in that one BRUV deployment that was removed from the analysis as stated in L164-165?

Reviewer #2: Well done and interesting manuscript. The figures and tables are tidy and well-presented. The information in the manuscript is pretty complete and easy to understand. I have made suggestions in the attached document that I hope improve the readability and clarity. In particular the Methods need some added information as it does not seem as polished as the other sections.

6. PLOS authors have the option to publish the peer review history of their article (what does this mean?). If published, this will include your full peer review and any attached files.

Reviewer #1: No

Reviewer #2: **Yes: **Dr Tiffany Sih

---

## [Author Response · Author response to Decision Letter 0]

31 Jan 2022

Response to reviewers’ comments

Reviewer #1

This is one of the few studies that documented patterns in the abundance and biomass of predatory fish assemblages in remote atoll seamount MPAs. The study used baited cameras to efficiently sample large areas of the lagoon and reef slope of the atoll, extending to the mid-mesophotic zone. The data presented are a valuable contribution to the coral reef ecological literature as previous studies of reef fish assemblages on remote atolls are mostly limited to euphotic depths. The ms is relatively well written but there is room for some improvement. I hope the general comments below will help improve the paper. Also listed below are minor comments, suggestions and corrections.

Reply: Thankyou. We have addressed your comments below that have greatly improved the manuscript.

General comments

1. The work highlighted the important roles of predatory fishes in the trophic ecology of reefs, the concerns about their dwindling numbers due to overfishing, and the importance of establishing baseline reference points for their spatial distribution (L39-49). However, it was difficult to appreciate the data presented in light of these points without showing some data on the overall fish assemblage. What proportion of the species richness, abundance and biomass of the overall fish assemblage in the atoll can be accounted for by predatory fishes? How do these proportions change with location and depth within the atoll? Do these proportions differ with other remote atolls in other regions that are open or closed to fishing? Addressing these questions may enhance the value of the ms. This seems fairly easy to do because the patterns presented here were simply extracted from the pooled fish assemblage data (L151-152).

Reply: We have provided an additional figure (Fig 2) along with the accompanying text to lines 271-276 to highlight the dominance of predatory fish abundance and biomass within the overall assemblage, and how this changes proportionally over depth and between locations. The entirety of Middleton Reef is protected in an IUCN II zone, so no legal fishing occurs on this reef. This is outlined on line 122.

2. Distinguishing patterns in the top predators from those of mesopredators seems useful and important to this ms. In many sections (e.g. L39, L58, L353-358, L399-405), the text seems to suggest a desire to highlight patterns in the abundance and biomass of top predators like sharks, large carangids and large groupers. There was also a desire to highlight the mesopredators (L376, L406). However, the ms did not explicitly identify which ones are top predators or mesopredators, for instance in Tables 1, 3, 5, and S1.

Reply: We have added trophic classifications to Table S1 to highlight which species we have considered apex from meso. This now clarifies the various existing and new mentions throughout the text. Where apex or meso is used in the text we have attempted to qualify which species we are specifically talking about (e.g. line 303-308).

3. The study highlighted significant differences in the structure of predatory fish assemblages between the upper- and mid-mesophotic zones and the lagoon (L333-334). However, the results did not really support this because partitioning into upper (30-60 m) and mid (60-90) mesophotic zones (L54) was not explicitly considered in the sampling design and data analysis (i.e. depth was treated as a continuous variable).

Reply: The spatially-balanced design did take into account the range of depths that cover these depth zones. However, as the reviewer highlights, we did not specifically classify depth into these zones for any analysis. Accordingly, we have removed reference to this depth partitioning to reduce any confusion. For example, the sentence on line 459 now reads “…across lagoonal to mesophotic shelf habitats.”

4. The depth range between about 11 to 29 m was not sampled in the south part of the atoll (Table 1). Was this a consequence of spatially balanced sampling that was weighted towards more complex habitat (L118)? Or was this simply a consequence of difficulties sampling that depth range in the south part of the atoll due to wave exposure? In any case, this gap in sampling should be clarified in the methods. More importantly, its potential implications for the conclusions must be discussed.

Reply: We have added further details on the sampling design and issues we faced during survey to lines 158-165 to clarify why we could not sample areas of the reef including the shallow reef margins in the south. We have also inserted a short sentence in the results on lines 292-294 to justify that this is unlikely to impact results due to similarities in accumulation curves.

Minor comments, suggestions and corrections

L18-20 – Sentence about Middleton Reef seems out of place here. It can be moved further down, around L24 where the authors described where the study was conducted. 

Reply: Moved to lines 23-24

L28 – 90% of the total fish biomass or just the total biomass of large predators? Needs clarification.

Reply: Total biomass of predatory fish. Has been updated on line 30.

L47 – Knowledge may be a better word to use here than understandings.

Reply: Changed to “knowledge” on line 49

L81 – key predatory species? 

Reply: Changed to “key predatory fish species” on line 105-106

L85 – Sentence needs improvement. Suggestion: “Bait was used to attract fish following methods that were approved by the Animal Ethics Committee…” 

Reply: Changed to “Bait was used to attract fish following methods that were approved by the University of Tasmania Animal Ethics Committee” on line 114-116

L104 – Specify that you are referring to habitats and faunal communities of Middleton Reef

Reply: Have inserted “…of Middleton Reef…” to line 137

L125 – austral (as opposed to boreal), not Astral

Reply: Changed on line 169

L126 – Odd to read numbers here instead of the authors

Reply: Have added “Langlois et al.” before numbers line 171. Happy to alter as per Editor advice

L131 – Must be camera lens, not field (and field of view is another matter) 

Reply: Yes, have altered text to “an estimated 8 m of the stereo BRUV to…” on line 185

L133 – higher order predators and mesopredators 

Reply: Have added “predators” to line 177

L135 – were instead of was 

Reply: Replaced with “were” on line 180

L147 – Needs correction and further explanation. Parameters a and b of length-weight models for each species are what can be sourced from Fishbase, not biomass calculations. Were length-weight models (in FL) available for all species? If not (or only available for TL), what was the procedure?

Reply: We have altered text on Lines 189-202 to correct this and provide further explanation around process. All fish (unless ray) were measured using FL.

L148 – spell out QA/QC (quality control and assurance?) 

Reply: Changed to “Quality control and assurance” on line 202-203

L151 – by dividing abundance AND biomass?

Reply: Changed to “or” as it is exclusive from each other on line 206

L154 – Suggest providing an idea of the number of times double-counting was suspected for these mobile predators, and the procedure taken to omit suspected repeats counts. 

Reply: The following text has been added to lines 210-212 to clarify this point: “Where double counting was suspected the individual was excluded from the dataset, which only occurred in the large tiger shark individuals recorded in the lagoon.”

L161 – of THE predatory fish assemblage…

Reply: Changed on line 223

L164-165 – Clarify that you are referring to abundance of the selected predatory fishes only. 

Reply: Inserted “predator” to indicate we are only talking about predatory fishes and not entire assemblage on line 236

L168 – Suggest modifying to “abundance and biomass structures of the predator assemblage across area (north, south and lagoon) as a fixed factor” 

Reply: changed on line 231-233

L200 – Check value for average abundance across all deployments stated in text against Table 1 

Reply: Thankyou for picking up this typo. Have updated SD text on line 280 to match table 1 values

L202-203 – Figure 2 does not convincingly show that sampling in the north and south areas adequately captured predator diversity. I tend to disagree that predatory species accumulation curves “largely reached asymptotes” in these regions. Instead, what Fig. 2 suggests is that the north and south areas have a higher species richness of predatory fish and that these areas were not as well-sampled as the lagoon. I think the original statement should be toned down and the higher species richness outside of the lagoon of Middleton Reef should be highlighted.

Reply: We have toned down this statement in line with the above comment. Lines 310-318 now reads: “The species accumulation curve for the predator fishes in the lagoon reached asymptote, indicating that sampling effort adequately captured diversity (Fig 3). However, the species accumulation curve for the predator fishes in the north and south areas suggest that although outside reef areas were not as well sampled, there was a higher predator richness outside of Middleton Reef lagoon (Fig 3). Importantly, both north and south areas exhibited similar accumulation patterns suggesting that, although there were issues accessing the shallow reef margins (i.e. 10-20 m) in the south, it is unlikely to overly impact results.” 

L223-226 – This sentence was a little confusing because it shifts its reference from total biomass to average biomass per deployment. Also, it should refer to Table 1 not S1.

Reply: We have changed text to be all average for abundance and biomass. We have restructured these sentences on lines 314-321. Also updated to S1 to Table 1 on lines 324 and 325.

L242, 246 – fish not fishes 

Reply: Changed to fish on lines 345 and 350

L251-255 – Check sentence construction. This is a very long sentence that is hard to read. 

Reply: We have completely rewritten this section of results to improve clarity (Lines 355-365).

L302 – GAMs not GAM’s 

Reply: Changed to “GAMs” on line 423

L304 – bohar not Bohar 

Reply: Removed when modified ms to use common names throughout as per Reviewer 2 comments

L306-310 – Again, a very long sentence that is difficult to understand. Suggest reconstructing or splitting. 

Reply: We have restructure both sections to reduce sentence length to improve clarity on lines 424-428.

L330 – five species THAT contributed to…

Reply: inserted “that” to line 454

L353 – remove semicolon 

Reply: deleted semicolon on line 503

L354 – Odd to read a number here instead of the authors [433]

Reply: Have inserted authors in text to line 505

L374 – Remove open parenthesis

Reply: removed parenthesis on line 527

L402-403 – E. daemelii is listed as near threatened (NT) which is NOT threatened, at least not yet.

Reply: Agree, this is not formally listed internationally, but is species identified as vulnerable in NSW. We have revised to text as follows “…high abundances of vulnerable species…” on line 538

L404-405 – I didn’t fully understand the latter part of this sentence. It seems to say that C. galapagensis are likely to be vulnerable even if adult distributions of the species are unknown. But the species is listed as least concern (LC) – Table S1

Reply: We have removed the latter part of this sentence to reduce confusion from line 541

Fig. 3 caption – Were the outliers removed from this plot all records in that one BRUV deployment that was removed from the analysis as stated in L164-165?

Reply: We have removed the sentence about a single outlier on line 228 to remove confusion as it was just individual values state in Fig 4 caption to were removed. We have also spelt this out in Fig 4 caption.

Reviewer #2

Well done and interesting manuscript. The figures and tables are tidy and well-presented. The information in the manuscript is pretty complete and easy to understand. I have made suggestions in the attached document that I hope improve the readability and clarity. In particular the Methods need some added information as it does not seem as polished as the other sections.

6. PLOS authors have the option to publish the peer review history of their article (what does this mean?). If published, this will include your full peer review and any attached files.

Do you want your identity to be public for this peer review? For information about this choice, including consent withdrawal, please see our Privacy Policy.

Reviewer #1: No

Reviewer #2: Yes: Dr Tiffany Sih

Review for PLOS One

Depth and benthic habitat influence shallow and mesophotic predatory fishes on a remote, high-latitude coral reef

Brown, Monk, Williams, Carroll, Harasti, Barrett

Spatial distribution and extent of predatory fish in shallow to mesophotic shelf environments

BRUVS lagoon and outer shelf habitats 0-100 m

Remote location subject to varied oceanographic processes

FYI, Check journal’s taxon naming specifications before publication, for Journal of Fish Biology the common name and authority/year need to be included at first mention of species. While some common names are included not all species in the text have common names mentioned, also be careful of capitalisation depending on if you are using common names or the Australian Standard Names (e.g. Fishes of Australia website for correct info). Would be helpful to include more common names as it could help internet searches to your publication and also is helpful to include in tables, etc for enhanced understanding. 

Reply: Thank you. We have used common names throughout based on Fishes of Australia. We have provided the scientific name at first use of common name. We have added both common and scientific names to Table S1 as well. There is no need to provide authority or year associated with PLoS guidelines.

Abstract: 

Line 20: Add “fish” in between “predatory populations”

Reply: Completed on line 20

Line 27: “…five predatory fish” makes it sound like you will list them. Not necessary here because of abstract word limits but I would suggest replacing “while” with “and” and starting a new sentence about Galapagos shark and Black cod or in some way splitting that long sentence. 

Reply: We have split the sentence following your suggestion on line 30

Introduction:

Line 28: be consistent with capital letters and if you ae using Australian Standard Names (both capitalised) or ‘common names’ are generally not capitalised. Fishes of Australia website has the correct ASN

Reply: Thank you for this insight. Names are now adjusted to common names based on Australian Standard Names.

Line 47: Baseline understanding (singular?)… is necessary 

Reply: Have altered to “Understanding” on line 51

Line 60-62: split the sentence/re-write [66-68] 

Reply: We have thoroughly re-written this paragraph to improve clarity (lines 55-97)

Materials and methods

Line 92: oceanographic influences (plural?) 

Reply: Changed to “influences” on line 124

Line 93: replace influence with strength or magnitude? 

Reply: Changed to “magnitude” on line 126

Line 101: add latitudinal degree and specify typical of tropical or shallow-water corals (vs deepwater coral latitudinal limits). “what is the lat limits of coral growth” (~ lat) of tropical. 

Reply: Have defined this on line 133-135

Line 105: “The reef structure morphology resembles a system that is modified by both hard coral formation and rubble accretion, which reflects seasonal variation…” This sentence could be improved.

Reply: Revised sentence structure to improve clarity on line 138-140

Line 119: “across depths from just below the surface to 100 m deep.” 

Reply: changed on line 154

Line 121: Source of existing bathymetry? Geoscience Australia or?

Reply: We have added “sourced from Geoscience Australia” to line 157-158

Line 125: austral summer

Reply: corrected on line 169

Paragraph beginning Line 118 or Line 123: add approx minimum space between deployments to maintain independence of replicates (on the map they look close, but that is probably just the size of the markers)

Reply: We have added “…with a minimum spacing of 250m between concurrent deployments.” to line 154-155

Line 130: Sardinops sagax

Reply: changed on line 175

Line 138: position of “.” Should be “GoPro .mp4”?

Reply: inserted on line 183

Line 140: “within an estimated 8 m field of view”

Reply: We have altered text in line with Reviewer 1 suggestion. Line 185 now reads: “All fish within an estimated 8 m of the stereo BRUV to the lowest classification possible were annotated”

Line 142: “and length measurements were taken for each individual encountered”

Reply: Altered on line 188

Line 143: “For teleost fishes, length was measured from the most anterior position (snout) to tail fork. Rays were measured across the widest portion of the disc. Other elasmobranchs were measured from snout to posterior tip on caudal fin.” Was this total length?

Reply: We have altered the text to improve clarify this point on lines 188-202

Line 145: “When fish were in large schools…” here you are referring to a school of one species or a school with more than one species? I would re-write this sentence to specify that you measured 20 individuals of the same species.

Reply: Correct, individual species, not mixed schools. We have altered the text to improve clarify this point on lines 191-194.

Line 154: either a comma is needed between the common and scientific names or move the parentheses?

Reply: inserted a comma as it is already within parentheses on line 209

Paragraph beginning Line 157: more information is needed like the number of habitat points per deployment or was this based on estimated percent cover? How many habitat categories and what were they? From what I can remember CATAMI just specified the naming convention/types/categories, but not specifically how to measure/quantify within a BRUVS field of view.

Reply: We have provided additional information on lines 216-221

Line 163 “data did not need to be transformed”? Not sure if untransformed is a word. Also, raw data does not necessarily need to be transformed but I would clarify if you used any tests (visual or statistical) to check the distribution of data, etc? Also, if this is the first mention of Primer, please include the software details here.

Reply: We have altered the text here to improve logic on line 225-226. We have also added citation for PRIMER on line 225 full details are provided on line 262-263

Line 164: One outlier… I would add more information here. Was this a replicate where no individuals were sampled?

Reply: We have deleted this sentence and provided more detail in caption of Fig 4 following Reviewer 1 comments.

Line 168: “across the fixed factor area”? Also this sentence is confusing how it is worded, perhaps move the clause “on the predator…and biomass structure” to the end of the sentence. Also, in this instance can “predator community assemblage abundance” be shortened to “predator relative abundance”?

Reply: we have reworded this sentence on line 230-233

Line 169: I would clarify that the SIMPER function is the “percentage similarity of….” And clarify if you used presence-absence or abundance data.

Reply: We have altered text on line 234 inline with this comment

Line 173: missing “transformed”

Reply: inserted on line 239

Line 174: “Vectors representing the most influential species/habitat variables were overlaid…”?

Reply: inserted on line 239-240

Paragraph beginning Line 176 could be re-written for clarity with more details. E.g., “To understand the relationships between taxa and measured habitat variables, a number of analyses were conducted. Species influential in distinguishing communities among areas around Middleton Reef (north, south, lagoon) from the SIMPER analysis were then used to test the relationships between habitat variables. Only species with sufficient abundance (MaxN > xx or n � xx ) were included.”

Reply: Have replaced start of paragraph you reviewers suggested text on lines 242-249

Line 182: split up sentence for clarity and include a citation for Tweedie distribution being an appropriate choice.

Reply: Split sentences and added citation for Tweedie on line 255

Line 184-185: Change “3” to “three”

Reply: changed on line 257

Line 192: here the use of ‘predatory’ seems like the species accumulation curve was being aggressive. Perhaps change to Predator species accumulation curve or Species accumulation curves of predatory species? Or add a ‘ after species. Line 202 and 204 same comment. Note this is just a suggestion but I think predator species sounds better than predatory species also throughout the results section.

Reply: we have altered text on lines 286-294 to say “Species accumulation curves of predator…”. We have changed to predator fish/species throughout where appropriate.

Line 215: This is not the correct common name for Pristipomoides filamentosus. It is easy to be confused because there are many names but most often Rosy Jobfish (ASN) or crimson jobfish are used, sometimes rosy snapper. This is to differentiate it from the other deepwater and eteline snappers (e.g. lavendar jobfish, ruby snapper, flame snapper and the half dozen species referred to as ‘red snapper’) ;-)

Reply: We have altered to Rosy Jobfish throughout.

Results

The results section is well-written. I only have a few comments.

Reply: Thankyou

Line 263 I think Pearson should be capitalised.

Reply: altered on line 380

Table 3: Not necessary but it might be nice to have some way of distinguishing the most similar/dissimilar, either with color, font differences (i.e. bold) or like a heat map. 

Reply: We have not altered table as we have highlighted the key species in the text on line 358-359 and 363.

Line 302: Remove the apostrophe after GAM, should be GAMs

Reply: removed on line 423

Line 317: In serranid the ‘s’ should be lowercase but for Serranidae the ‘s’ is capitalised.

Reply: We have altered throughout

Table 5: Great table!

Reply: Thankyou

Discussion

Overall I found the Discussion well-written and interesting. I only have a few comments. 

Reply: Thankyou

Paragraph beginning Line 339: This is a big paragraph and might benefit from splitting. Perhaps around “Complex habitat…” how you do it is up to you.

Reply: We have split paragraph at “Previous studies…” on line 480

Line 372: change fishing to “fishing effort”

Reply: inserted “effort” to line 500

Line 381: This first/topic sentence lacks a little oomph. I think it can be fixed by switching the order of words or a different word than concide. It does not match with the direction of the paragraph. 

Reply: We agree. We have deleted the entire first sentence on line 510-511 as we agree it didn’t fit with remaining direction of the paragraph.

Line 391: I would write out East Australian Current (EAC) since it is the first mention in the Discussion. 

Reply: Spelt out in full on line 520-521

Conclusion

Line 402-403: This sentence could be re-worded/re-structured for clarity

Reply: We have worded the sentence on line 536-537 for clarity

Line 407: beter segue between sentences? Or the sentence beginning with “Declines…” could be moved to the beginning of the paragraph

Reply: We have moved the sentence and next that were originally on lines 543-546 to start of paragraph (lines 529-532)

---

## [Editor Report · Decision Letter 1]

23 Feb 2022

Depth and benthic habitat influence shallow and mesophotic predatory fishes on a remote, high-latitude coral reef

PONE-D-21-27786R1

Dear Dr. Monk,

We’re pleased to inform you that your manuscript has been judged scientifically suitable for publication and will be formally accepted for publication once it meets all outstanding technical requirements.

Kind regards,

Fraser Andrew Januchowski-Hartley, Ph.D.

Academic Editor

PLOS ONE
---

## [Editor Report · Acceptance letter]

11 Mar 2022

PONE-D-21-27786R1 

Depth and benthic habitat influence shallow and mesophotic predatory fishes on a remote, high-latitude coral reef  

Dear Dr. Monk:

I'm pleased to inform you that your manuscript has been deemed suitable for publication in PLOS ONE. Congratulations! Your manuscript is now with our production department. 

Kind regards, 

on behalf of

Dr. Fraser Andrew Januchowski-Hartley 

Academic Editor

PLOS ONE